# Analysis of *GEN1* as a Breast Cancer Susceptibility Gene in Polish Women

**DOI:** 10.3390/ijms26135991

**Published:** 2025-06-22

**Authors:** Katarzyna Gliniewicz, Dominika Wokołorczyk, Wojciech Kluźniak, Klaudia Stempa, Tomasz Huzarski, Helena Rudnicka, Anna Jakubowska, Marek Szwiec, Joanna Jarkiewicz-Tretyn, Magdalena Cechowska, Paweł Domagała, Tadeusz Dębniak, Marcin Lener, Jacek Gronwald, Jan Lubiński, Steven A. Narod, Mohammad R. Akbari, Cezary Cybulski

**Affiliations:** 1International Hereditary Cancer Center, Department of Genetics and Pathology, Pomeranian Medical University in Szczecin, 71-252 Szczecin, Poland; katarzyna.gliniewicz@pum.edu.pl (K.G.); dominikawok@gmail.com (D.W.); wojciech.kluzniak@pum.edu.pl (W.K.); klaudia.stempa@pum.edu.pl (K.S.); tomasz.huzarski@pum.edu.pl (T.H.); anna.jakubowska@pum.edu.pl (A.J.); tadeusz.debniak@pum.edu.pl (T.D.); marcin.lener@pum.edu.pl (M.L.); jacek.gronwald@pum.edu.pl (J.G.); jan.lubinski@pum.edu.pl (J.L.); 2Department of Clinical Genetics and Pathology, University of Zielona Góra, 65-046 Zielona Góra, Poland; 3Independent Laboratory of Molecular Biology and Genetic Diagnostics, Pomeranian Medical University in Szczecin, 71-252 Szczecin, Poland; 4Department of Surgery and Oncology, University of Zielona Góra, 65-046 Zielona Góra, Poland; szwiec72@gmail.com; 5Cancer Genetics Laboratory, 87-100 Toruń, Poland; jarkiewicztretyn@poczta.onet.pl (J.J.-T.); mcechowska@gmail.com (M.C.); 6Women’s College Research Institute, Women’s College Hospital, Toronto, ON M5S 1B2, Canada; steven.narod@wchospital.ca (S.A.N.); mohammad.akbari@utoronto.ca (M.R.A.); 7Dalla Lana School of Public Health, University of Toronto, Toronto, ON M5T 3M7, Canada

**Keywords:** *GEN1*, germline mutations, susceptibility, cancer

## Abstract

*GEN1* is implicated in DNA damage repair, as are several other breast cancer susceptibility genes, and is included in several comprehensive next-generation sequencing (NGS) testing panels. To investigate the possible association of *GEN1* variants with breast cancer risk, we sequenced this gene in 617 Polish women with hereditary breast cancer (HBC) and 300 Polish cancer-free controls. No protein-truncating variants were detected in the conserved part of *GEN1* (first 480 codons). Two frameshift variants were detected in the last exon of *GEN1*: c.2515_2519delAAGTT (p.Lys839Glufs*2) and c.1929_1932delAAAG (p.Lys645Cysfs*29). The p.Lys839Glufs*2 variant was detected in 21.1% of 617 HBC cases and 18.4% of 300 controls (*p* = 0.38). The p.Lys645Cysfs*29 variant was rare, seen in 0.6% of 617 HBC cases and 0.3% of 300 controls. The variant was then detected in 38 (0.24%) of 15,930 unselected breast cancer cases and 8 (0.17%) of 4702 cancer-free female controls from Poland (OR = 1.40, *p* = 0.49). Clinical characteristics of breast tumors in the 38 carriers of p.Lys645Cysfs*29 and 15,892 non-carriers were similar. Survival was similar among variant carriers and non-carriers (the age-adjusted HR = 0.87, *p* = 0.76). The wild-type *GEN1* allele was retained in all five breast cancers of carriers of p.Lys645Cysfs*29. No cancer type was more frequent in the relatives of 35 p.Lys645Cysfs*29 variant carriers compared to the relatives of 14,592 non-carriers. We conclude that *GEN1* is unlikely to be a high or moderate-risk breast cancer susceptibility gene. Our study has clinical implications for genetic counseling and suggests that *GEN1* changes should be reclassified as variants of uncertain significance (VUS) when they are detected in clinical testing panels.

## 1. Introduction

Breast cancer is one of the major causes of morbidity and mortality from malignant tumors in women worldwide. Inherited background plays an important role in the etiology of this disease [1]. According to the 2022 GLOBOCAN report, breast cancer is the most commonly diagnosed cancer in women worldwide, with approximately 2.3 million new cases and over 670,000 deaths annually [2]. About 15–20% of breast cancer patients report a positive history of breast cancer in one or more relatives, including families with a strong clustering of breast cancer (hereditary breast cancer; HBC). Epidemiologic data suggest that mutations in highly and moderately penetrant genes are responsible for about 10–15% of all breast cancer cases [3]. Based on mutation frequencies and cancer risk, the two most important susceptibility genes for breast cancer are *BRCA1* and *BRCA2* [4]. Other genes for which the evidence of the association with breast cancer is strong include *PALB2*, *CHEK2*, *ATM*, *BARD1*, *RAD51C*, *RAD51D*, *TP53*, *PTEN*, *NF1*, *STK11*, and *CDH1* [5,6,7,8,9]. Clinical genetic testing for mutations in these genes is useful in risk prediction, cancer prevention, surveillance in unaffected individuals, or treatment of cancer patients [10,11,12,13]. Most recently, based on cancer-specific mortality/morbidity benefit, the ESMO Precision Oncology Working Group recommended that the clinical testing panel should include eight genes: *BRCA1*, *BRCA2*, *PALB2*, *RAD51C*, *RAD51D*, *BRIP1*, and *TP53* [14]. It is also important that several of these genes predispose to ovarian cancer, and preventive salpingo-oophorectomy is therefore warranted [15].

Most breast cancer susceptibility genes are involved in the damage repair signaling pathway, i.e., including *BRCA1*, *BRCA2*, *PALB2*, *CHEK2*, *ATM*, *BARD1*, *RAD51C*, and *RAD51D* [16,17,18]. The major function of this pathway is the repair of double-strand DNA breaks (caused by mutagenic factors, i.e., chemicals and ionizing radiation) via homologous recombination using the homologous intact chromosome as a template for repair [19,20,21,22]. During this process, DNA intermediates, known as Holliday junctions (HJs), are formed in the course of DNA repair, and their resolution is needed for proper chromosome segregation. Once repair is complete, the junctions are broken, and two double-stranded DNA molecules are formed [23,24,25,26,27,28].

*GEN1* (OMIM Number: 612449) is an important gene in the DNA repair pathway. It is located on chromosome 2p24.2. The coding sequence of the human *GEN1* is in 13 exons. It encodes a monomeric 103-kDa protein of 908 amino acids in length [29]. The GEN1 protein contains three functional motifs: N-terminal and internal xeroderma pigmentosum group G nuclease domains (the XPG-N and XPG-I), followed by a helix-hairpin-helix domain [30,31,32,33]. It encodes for a protein, which is a member of the Rad2/xeroderma pigmentosum group G (Rad2/XPG) family of endonucleases, and plays a key role in resolving structural DNA junctions (DNA duplexes) such as Holliday junctions (HJs). GEN1 dimerizes upon binding to HJs. GEN1 introduces notches in symmetrically bound DNA strands, leading to the separation and formation of products in the form of nicked DNA duplexes (HJs). Therefore, the GEN1 protein is critical in resolving Holliday junctions by introducing symmetrically related cuts across the junction point to produce nicked duplex products in which the nicks can be readily ligated during DNA repair by homologous recombination [25]. GEN1 also shows activity against structures such as 5′-flaps and replication forks [34]. In addition, it has been suggested that it maintains centrosome integrity [35,36,37]. Based on its critical function in DNA repair and the maintenance of genomic stability through the resolution of Holliday junctions, we considered *GEN1* to be a candidate breast cancer susceptibility gene. However, there is currently a lack of epidemiological data that supports the association of germline variants in *GEN1* with increased cancer risk [14,25].

In this study, we investigated the potential contribution of *GEN1* to breast cancer susceptibility. First, we screened the entire coding sequence of the *GEN1* gene in 617 Polish patients with hereditary breast cancer and 300 Polish population controls by exome sequencing (discovery sequencing phase) to search for putative loss-of-function mutations (protein-truncating variants). We then conducted an association study of a rare recurrent frameshift variant c.1929_1932delAAAG (p.Lys645Cysfs*29) (detected by sequencing in the 617 HBC patients) with breast cancer risk in 15,930 unselected Polish women with breast cancer and 4702 cancer-free female controls (validation genotyping phase). Further, we compared the clinical characteristics of breast cancers in carriers of the c.1929_1932delAAAG allele and non-carriers. We also conducted a loss of heterozygosity (LOH) analysis at the *GEN1* locus in breast cancer tissues from five patients with the frameshift variant of *GEN1*. To study if the c.1929_1932delAAAG deletion confers susceptibility to cancers at other sites, we analyzed the pedigrees of breast cancer patients with and without the variant allele.

## 2. Results

We identified two putative loss-of-function variants of *GEN1* among 617 Polish breast cancer patients with HBC and 300 controls by exome sequencing. Both variants were small intragenic frameshift deletions located in the last exon of *GEN1* (Table 1).

One was a common truncating variant (c.2515_2519delAAGTT, p.Lys839Glufs*2), which in the homozygous or heterozygous state was present in 21.1% of 616 patients with HBC and 18.4% of 299 controls (*p* = 0.38). In the homozygous state, the deletion was less frequent in HBC cases (0.8%) than in controls (1.3%), but the difference was not significant (*p* = 0.48). In the heterozygous state, it was observed in 20.3% of HBC cases and 17.3% of controls (*p* = 0.28). The c.2515_2519delAAGTT deletion was present with similar frequency in HBC cases and controls and therefore was not studied further. The other variant was a recurrent frameshift deletion c.1929_1932delAAAG (p.Lys645Cysfs*29). It was more common in HBC cases than in controls (0.6% vs. 0.3%), but the numbers were small and the difference was not statistically significant (*p* = 1.00).

The p.Lys645Cysfs*29 variant was then genotyped in 15,930 women with unselected breast cancer and 4702 cancer-free female controls. It was observed in 38 (0.24%) of unselected breast cancer cases and in 8 (0.17%) of unaffected controls (OR = 1.40, *p* = 0.49). Among the unselected cases, the variant was found in seven of 2307 patients (0.30%) with familial breast cancer (OR = 1.79, *p* = 0.39). However, the variant is rare, and the 95% CI is wide both for unselected cases (0.66–3.01) and familial cases (0.65–4.93); therefore, due to limited power, our study does not exclude a modest increase in breast cancer risk associated with the variant, but it suggests that it is unlikely that the variant confers high or moderate risk of breast cancer. The mean age of diagnosis was 56.8 years among variant carriers and 56.0 years in non-carriers (*p* = 0.64, *t*-test). The variant frequency was 9 of 5724 in women diagnosed at age 50 or below, compared to 29 of 10,207 patients diagnosed above age 50 (*p* = 0.16). The frequencies of the deletion by age and by family history of breast cancer are presented in Table 2.

We then compared the clinical characteristics of the breast cancer patients with and without the *GEN1* c.1929_1932delAAAG variant. There were no statistically significant differences in the clinical characteristics of the breast cancers between variant carriers and non-carriers by age at diagnosis, estrogen receptor (ER), progesterone receptor (PR), human epidermal growth factor receptor 2 (HER2) status, tumor histology, size, and lymph node involvement. However, the statistical power of this analysis was limited. A detailed comparison is shown in Table 3.

Data on all-cause survival were available for 12,067 breast cancer patients. The median follow-up period was 67 months (range 13 to 169) in 30 carriers of c.1929_1932delAAAG and 60 months (range 1 to 223) in 12,037 non-carriers (*p* = 0.82, Mann-Whitney test). During the follow-up time, we observed five deaths among the 30 (16.7%) carriers of the c.1929_1932delAAAG deletion and 2036 (16.9%) in 12,037 non-carriers (*p* = 0.97). The 5-year survival was 86% in carriers and 84% in non-carriers. The 10-year survival was 77% in carriers and 75% in non-carriers. The age-adjusted HR for mortality given the *GEN1* c.1929_1932delAAAG variant was 0.87 (95% CI 0.36 to 2.09; *p* = 0.76; Cox regression analysis). To determine if there is an excess of malignancies at other sites in first- and second-degree relatives, we reviewed the pedigrees of patients with and without c.1929_1932delAAAG. No cancer was more frequent in the relatives of the deletion carriers compared to the relatives of non-carriers (Table 4).

We also analyzed the tumor DNA samples from breast cancer tissues of five carriers with a *GEN1* p.Lys645Cysfs*29 small deletion for loss of heterozygosity at the *GEN1* locus. To investigate if the wild-type allele is lost in cancer tissues, we calculated the ratio between the peak height of the first deleted nucleotide-nucleotide G corresponding to the wild-type allele (marked by “a” on Figure 1) to the height of the control peak corresponding to the closest nucleotide G located in the normal sequence, 3 bp 5′ to the deletion (marked by “b” on Figure 1). For five germline DNA samples of heterozygous *GEN1* variant carriers (controls), the median peak ratio a/b was 0.46 (range 0.43 to 0.51). For all breast cancer tissues from deletion carriers, the ratios were similar to those of the controls heterozygous for the variant (all ratios > 0.43), suggesting that the wild-type *GEN1* allele was retained in all cancers (Figure 1). This is consistent with the observation that the signal from the first deleted nucleotide corresponding to the wild-type allele (nucleotide G, black peak marked by “a” on Figure 1) is strong in all breast cancers compared to that of nucleotide T specific for the deletion (red peak marked by “a” on Figure 1), and in the case of LOH, the black peak of the first deleted nucleotide G should be minimal, almost completely lost.

## 3. Discussion

The *GEN1* gene is implicated in DNA damage repair, as are several high- or moderate-risk breast cancer susceptibility genes, i.e., *BRCA1*, *BRCA2*, *PALB2*, *CHEK2*, *ATM*, and *BRIP1* [16,17,18]. Somatic frameshift mutations in *GEN1* have been identified in breast cancer cell lines and primary tumors [38,39]. Therefore, we sought to ask if germline variants in *GEN1* predispose to breast cancer in Poland. We detected two recurrent frameshift variants (p.Lys645Cysfs*29 and p.Lys839Glufs*2) by sequencing the entire coding sequence of the gene of 617 Polish HBC patients and 300 cancer-free controls. One was a common small deletion, c.2515_2519delAAGTT (p.Lys839Glufs*2), that removes 9% of the 3′-end of its protein sequence. This deletion was present with a similar frequency (about 20%) in Polish HBC cases and controls, indicating that it does not confer an increased risk of breast cancer. Previously, this deletion was not associated with breast cancer in the UK [40]. In that study, it was detected with the same frequency (4%) in 1072 UK cases of breast cancer and 1050 controls. Another study from Russia reported that the p.Lys839Glufs*2 mutation in the homozygous state was overrepresented in 360 women with bilateral breast cancer compared to 1305 controls (3.1% vs. 1.4%, OR = 2.3, *p* = 0.03), but this trend was not observed for women with unilateral breast cancer (1.2% of 1851 patients vs. 1.4% of 1305 controls). In our study, neither homozygotes nor heterozygotes were associated with an increased risk of breast cancer, and we observed no c.2515_2519delAAGTT homozygotes among 44 Polish women with bilateral breast cancer. Therefore, the aggregate evidence suggests that the c.2515_2519delAAGTT deletion is not implicated in the etiology of breast cancer.

The other frameshift variant detected in Polish HBC cases was c.1929_1932delAAAG. This variant creates a premature translational stop codon (p.Lys645Cysfs*29) in *GEN1*. It is not anticipated to cause nonsense-mediated decay but is expected to disrupt the last 264 amino acids of GEN1. Experimental studies and prediction algorithms are not available, and its functional significance is unknown. The deletion is very rare in population databases (i.e., gnomAD; frequency 0.01%). However, the variant appeared more frequently in our Polish cohort (0.2%) and was overrepresented in HBC cases compared to controls (albeit the numbers were small), so we decided to explore if it is implicated in the pathogenesis of breast cancer. We approached this question in several ways, including a large case-control association analysis of 15,930 cases and 4702 controls and a pedigree study. In addition, we asked if breast tumors with and without this variant were different in their clinical characteristics and survival among patients. LOH is a signature for most cancer syndromes with homologous repair deficiency, so we also evaluated the loss of heterozygosity at the *GEN1* locus in breast cancer tissues from five carriers of c.1929_1932delAAAG. In neither of these approaches did we see evidence of cancer predisposition for carriers of a *GEN1* variant. This suggests that the c.1929_1932delAAAG allele is not pathogenic for breast cancer, but a modest increase in breast cancer risk associated with the variant cannot be excluded based on the odds ratio and confidence limits of 1.4 and 95% CI of 0.66–3.01 seen in our study.

Holliday junction resolution is essential for chromosome segregation at meiosis and the repair of stalled/collapsed replication forks in mitotic cells [21]. All organisms possess nucleases that promote HJ resolution. GEN1, a member of the Rad2/XPG nuclease family, was isolated recently from human cells and shown to promote HJ resolution in vitro and in vivo. GEN1 protein contains the XPG-N, XPG-I, and helix–hairpin–helix domains essential for nuclease activity and a C-terminal tail [41]. The XPG and helix–hairpin–helix domains are localized in the highly conserved region of *GEN1* over the first 480 amino acids [25]. We and Turnbull et al. [40] did not detect any truncating mutations in this conserved region of *GEN1* (the first 480 codons). Protein-truncating mutations in the C-terminal tail, such as variants p.Lys839Glufs*2 and p.Lys645Cysfs*29, were not associated with breast cancer susceptibility. This is consistent with the findings that a truncated form of GEN1, lacking the C-terminal, is sufficient for GEN1 nuclease activity [25]. In more detail, functional studies indicate that the catalytic activity of GEN1 is confined to its highly conserved N-terminal region, which includes the XPG-N and XPG-I domains. Ip et al. demonstrated that the C-terminal region of GEN1 is dispensable for its Holliday junction resolvase function [25], and Rass et al. confirmed that truncations in this non-catalytic C-terminal domain do not compromise enzymatic activity [41]. Therefore, truncating variants such as i.e., p.Lys645Cysfs*29, which affect only the C-terminal tail, are unlikely to disrupt GEN1’s core DNA repair function. However, the role of p.Lys645Cysfs*29 on GEN1 function has not been explored, and it would be important to perform functional studies to validate the impact of p.Lys645Cysfs*29 (and other *GEN1* variants) on protein function. 

Turnbull et al. found no evidence for the association of six SNPs tagging common *GEN1* variants with breast cancer risk in a study of 3750 cases and 4907 controls [40]. Most recently, the Breast Cancer Association Consortium (BCAC) used a panel of 34 putative susceptibility genes to perform sequencing on samples from 60,466 women with breast cancer and 53,461 controls and identified only 31 carriers of *GEN1* truncating mutations among cases and 41 in controls (OR = 0.66, *p* = 0.18), but the localization of the truncating mutations was not provided [42]. In this BCAC study, rare missense variants in *GEN1* were also not associated with an increased risk of breast cancer (OR = 1.05, *p* = 0.25). However, *GEN1* truncating variants are uncommon, and further epidemiological and functional studies are needed to explore if *GEN1* mutations may confer increased breast cancer risk.

One strength of our study is the genetic homogeneity of Poland, which is populated by ethnic Poles, and we restricted our analyses to individuals who self-reported themselves as Poles (of note, both cases and controls were enrolled before the full-scale Russian-Ukrainian war in 2021, when Poland accepted above 1 million refugees from Ukraine). We don’t observe significant differences in the allele frequencies between different regions of Poland, i.e., for *BRCA1*, *PALB2*, *CHEK2*, *NBN*, *RECQL*, and also for the *GEN1* founder p.Lys645Cysfs*29 variant, neither in breast cancer cases nor in controls (for *GEN1* data, see Appendix A). In most breast cancer susceptibility genes, Polish common founder mutations are present, including *BRCA1*, *BRCA2*, *PALB2*, and *CHEK2*, which constitute the majority (>80%) of all mutations detected in these genes in Polish women with breast cancer [43].

We studied individuals from a genetically homogenous population, so we searched for recurrent (founder) truncating variants by sequencing 617 unrelated Polish HBC families. Using binominal distribution, we estimate that if an allele is present in 3 of the 1000 Polish HBC families in a given gene, we have a power of 55% to detect this allele in two or more families. If the allele is present in 6 of 1000 or more Polish HBC families, we have a power of above 88% to detect this allele in two or more families. In fact, we identified two recurrent variants of *GEN1*, one common allele (p.Lys839Glufs*2) detected in 21% of Polish HBC families and one rare variant (p.Lys645Cysfs*29) seen in 0.6% of Polish HBC families. Therefore, given the analysis of the power to detect *GEN1* variants in our discovery sequencing phase by binominal distribution (among 617 Polish HBC families), a large-scale multicenter association study in our genotyping validation phase, which included approximately 16,000 unselected breast cancer cases and 5000 controls from all over Poland, and a genetic homogeneity of the Polish population, we consider that we have sufficient data and power to detect possible associations between *GEN1* and breast cancer susceptibility. 

Testing for germline mutations by next-generation sequencing is important for breast cancer prevention, surveillance in unaffected individuals, and for managing cancer patients. Clinical testing panels comprise genes with strong, limited, or unknown associations with breast cancer and also include genes for cancer susceptibility at other sites. The current multigene tests include focused, guideline-based, and comprehensive panels. It is challenging to decide which panel to use in a particular clinical situation, i.e., for unaffected women with a positive cancer family history, women with breast cancer, and patients with metastatic disease, to consider the advantages and limitations of expanded genetic testing. In particular, this question applies to comprehensive testing panels that include genes associated with susceptibility to many cancers (i.e., breast, ovarian, colon, and prostate cancers). Although the association between *GEN1* with cancer risk is not established, this gene is included in the testing panels (i.e., the Fulgent Full Comprehensive Cancer Panel; and the Invitae DNA Damage Repair Panel). In the cases that a *GEN1* variant is found using the comprehensive testing panel, the variant carriers will want to know to what extent they are at elevated cancer risk. In this context, our study has clinical implications for genetic counseling and suggests reclassification of *GEN1* changes as variants of uncertain significance (VUS) when they are detected in clinical testing panels.

## 4. Materials and Methods

### 4.1. Participants

We used a sequential approach. First, we conducted full gene screening of 617 unrelated breast cancer patients with HBC and 300 controls (sequencing discovery phase). *GEN1* sequencing data were derived from a cohort of all 617 Polish women with breast cancer from HBC families who had previously been screened using exome sequencing (WES) by our group. To look for high- and moderate-risk variants/genes, we selected 617 Polish HBC families with a strong clustering of breast cancer (on average 3.4 breast cancer cases per family). In the discovery phase as a reference, we used 300 cancer-free Polish population controls who were screened by exome sequencing previously (as a consequence of our study “Polinome”, performed to obtain 300 Polish exomes from cancer-free individuals that may be used as a reference for the interpretation of pathogenicity of germline variants detected in Polish cancer patients). As the studied variants may be rare, in order to maximize the number of controls, we used all available 300 cancer-free Polish individuals who had exome sequencing, both women (n = 155) and men (n = 145). We saw no significant difference in the studied *GEN1* variant frequencies (for both truncating *GEN1* variants) by sex among the 300 controls.

In our second phase, we conducted a large case-control association study (the validation genotyping phase) to evaluate the contribution of a recurrent *GEN1* truncating variant to breast cancer susceptibility, which was identified in the discovery sequencing phase. The variant was rare (seen with a frequency <1% in 617 HBC cases in the discovery phase), so we genotyped all available unselected breast cancer cases (n = 15,930) and cancer-free female controls (n = 4702). Patients and controls used in the validation genotyping phase were recruited from multiple centers across Poland. Of note, only Polish cases and controls were enrolled in our study (all enrolled individuals reported that all their grandparents were of Polish ethnicity); therefore, the most important criterion of an association study, is that both cases and controls must be of the same ethnicity, is fulfilled. In addition, we saw no difference in the *GEN1* variant frequency by geographical region of Poland (Appendix A). All case and control series are described below. The flowchart showing the structure of our sequential approach is shown in Figure 2.

#### 4.1.1. Hereditary Breast Cancer Cases

This case series was used in the discovery sequencing phase of the study to identify mutations in *GEN1* in Polish women with breast cancer and HBC by sequencing. The series included 617 unrelated women with breast cancer from 617 families with strong clustering of breast cancer. The mean age of breast cancer diagnosis among the 617 probands was 46 years (range 28 to 76 years). A total of 160 women were from families with at least four or more women with breast cancer, 378 women were from families with three cases of breast cancer, and 79 women were from families with two affected (at least one bilateral and/or early onset breast cancer below the age of 50). The mean number of breast cancers per family was 3.4. All probands were Polish women. The families were selected from 3519 families registered at the Hereditary Cancer Center in Szczecin based on age of diagnosis, the larger number of breast cancer diagnoses with lower age of onset among relatives, and whether they tested negative for a panel of 17 Polish founder mutations of *BRCA1*, *BRCA2*, *CHEK2*, *PALB2*, *NBN*, and *RECQL* [44].

#### 4.1.2. Unselected Breast Cancer Cases

The group of unselected cases consisted of 15,930 prospectively ascertained cases of invasive breast cancer, diagnosed from 1996 to 2013 at 18 different hospitals in Poland (mean age of diagnosis 56 years, range 18–94). This series was used in the validation genotyping phase of the study. All women at the participating centers who received a primary diagnosis of invasive breast cancer were eligible. Individuals with ductal carcinoma in situ (DCIS) were included. Patients with intralobular carcinoma in situ (LCIS) were excluded. The patient participation rate was 76%. Patients were unselected for family history. All women tested negative for the three most common Polish mutations in *BRCA1* (C61G, 4153delA, or 5382insC). Information on the clinical characteristics of breast tumors was recorded through a review of medical data. Family history included all cases of any cancer in first- and second-degree relatives and their ages of diagnosis. Cancer family history was available for 14,592 of 15,930 (92%) of cases. Of the 14,592 unselected cases, 2307 patients (16%) reported breast cancer in one or more first- or second-degree relatives (familial breast cancer cases from unselected series). Survival data (all-cause mortality) were obtained (alive or dead and the date of death if deceased) from the Polish Ministry of the Interior and Administration in July 2014.

#### 4.1.3. Controls

We included two groups of non-overlapping controls. The first series consisted of 300 Polish cancer-free adults, 155 women (age range 40–84; mean 56.9) and 145 men (age range 45–89; mean 62.1), who underwent exome sequencing in our previous study and were used as a reference in the discovery sequencing phase of the study and are described in detail previously [45]. In brief, these adults were selected randomly from a registry of patients who participated in the population-based study of 1.5 million residents of West Pomerania, based on the fact that they were cancer-free and reported negative cancer family history in first-degree relatives [46]. This series was used to estimate frequencies of *GEN1* variants in the underlying Polish population and was applied as a reference to assess the association of *GEN1* variants (detected by sequencing) with hereditary breast cancer.

The second control group was recruited from four sources and included 4702 cancer-free women aged 20 to 94 years (mean age, 53.0), it was used as a reference in the validation genotyping phase of our study, and it is described in detail elsewhere [47]. In brief, it included four series: 959 female residents of the Szczecin area matched by age (range 24 to 84 years) and place of residence to a series of patients with incident breast cancer diagnosed in Szczecin between 1996 and 2004; 1717 unselected women (aged 32 to 72 years) who undergone breast ultrasound or mammography between 2009 and 2011 at eight Polish centers (women with breast cancer, and women with a family history of breast cancer were excluded from this series); 1036 unselected females (aged 20 to 94 years) randomly selected from the lists of primary care doctors in the Opole area in 2012 and 2013; 990 Polish women (50 to 66 years), who donated blood samples between 2007 and 2010 in Białystok, Łodź, and Szczecin during population colonoscopy screening program for colorectal cancer. This control group was used to analyze the association of a rare Polish founder frameshift variant c.1929_1932delAAAG of *GEN1* with breast cancer.

All individuals signed informed consent for genetic testing of their DNA samples. All cases and controls were of Polish ethnicity. The Ethics Committee of Pomeranian Medical University in Szczecin approved the study (IRB No. KB-0012/97/17).

### 4.2. Methods

#### 4.2.1. Sequencing of GEN1

We analyzed the entire coding sequence of *GEN1* from the exome sequencing data of 617 women with hereditary breast cancer and 300 controls. Germline DNA was isolated using standard methods from peripheral blood leukocytes. The Agilent SureSelect human exome kit (V6) was used for capturing target regions. The kit captures 64 Mbp (2.2%) of the human genome and covers coding exons in CCDS and RefSeq databases, as well as exons annotated by the GENCODE project. This includes ~205,000 exons in ~35,000 genes, including protein-coding and non-coding RNA genes.

The captured regions for each sample were barcoded, and every 16 samples were pooled and used for paired-end sequencing for 150 cycles (generating 150 bp reads) on a high-throughput sequencing cartridge of Illumina NextSeq 500.

The sequence reads for each exome sequence was aligned to the reference sequence of the human genome using the Burrows–Wheeler Aligner version 0.7.17. The mean depth of coverage was approximately 100× (range 52× to 154×). On average, 97.4% (range 91.2–99.1%) of the CCDS exons were covered at 20× depth of coverage or higher and were used for variant calling. For *GEN1*, on average, 99.8% of its coding exons (exons 2-14, NM_182625.4) were covered at 20× or more.

The Picard package (http://picard.sourceforge.net) was used to convert the SAM files to BAM format and sort and index the BAM files. In the next step, all the unmapped reads aligned to more than one human genome region, and all duplicate reads were filtered out from the BAM file using the Genome Analysis Toolkit (GATK) package version 4.0. Exact duplicate reads were collapsed to avoid inflated coverage. The HaplotypeCaller module of the GATK package was used for calling both SNPs and indels. Regions with at least 20-fold depth of coverage were used for calling variants, and a different nucleotide from the reference sequence seen in at least 25% of the reads aligned to a given position was called a variant. The SNP & Variation Suite (GoldenHelix Inc., Bozeman, MT, USA) was used for annotating called variants and to determine the effect of *GEN1* variants on the coding protein. We searched for loss-of-function variants, including frameshift indels, nonsense mutations, splicing site variants, and start codon loss mutations.

#### 4.2.2. Genotyping

Genomic DNA was isolated from 5 to 10 mL of peripheral blood by the non-enzymatic method [48]. The c.1929_1932delAAAG (p.Lys645Cysfs*29) variant was genotyped using a TaqMan assay (Thermo Fisher Scientific, Waltham, MA, USA) on a LightCycler Real-Time PCR 480 System (Roche Life Science, Mannheim, Germany) using probes, [HEX]ACACTGCAAACATAAAGAAAGT[BHQ1] and [6FAM]TACACTGCAAACATAAAGTGTT[BHQ1], and primers, forward—AAAATATCCCAGAACAACTGTCCTG and reverse—CTTCAGGACTAATCCCATCAGAATC. Real-time PCR conditions are described previously [40]. All variants were confirmed by Sanger sequencing. Sequencing reactions were performed using a BigDye Terminator v3.1 Cycle Sequencing Kit (Thermo Fisher Scientific, Waltham, MA, USA) according to the manufacturer’s protocol using forward primer CCAGAACAACTGTCCTGTGA and reverse primer TGCTCTTCAGGACTAATCCCA. Sequencing products were analyzed on an ABI Prism 3100 Genetic Analyzer (Thermo Fisher Scientific).

#### 4.2.3. Loss of Heterozygosity Analysis

Formalin-fixed paraffin-embedded (FFPE) breast cancer tissue samples from six *GEN1* mutation carriers were available in the Department of Pathology of Pomeranian Medical University in Szczecin. A pathological review of these samples was conducted by a pathologist. Tissue samples of good quality were available from five patients. So, we were able to obtain tumor samples and perform LOH for two *GEN1* variant carriers identified by WES among 617 patients with HBC and three carriers identified by genotyping of a series of 15,930 unselected breast cancer cases. Loss of heterozygosity (LOH) analysis at the *GEN1* locus was performed in DNA from micro-dissected tumors from five *GEN1* variant-positive women (p.Lys645Cysfs*29) as described previously [49] with some modification: (1) DNA was isolated with QIAamp DNA FFPE Tissue Kit (from QIAGEN, Hilden, Germany); (2) LOH was analyzed by direct Sanger sequencing of a 92 bp DNA fragment containing the p.Lys645Cysfs*29 variant (forward primer CCAGAACAACTGTCCTGTGA and reverse primer TGCTCTTCAGGACTAATCCCA). In brief, the pathologist associated with the study (P.D.) chose fields of tissue containing >80% of breast cancer cells on H&E (Hematoxylin and Eosin) stained slides using a light microscope. Then, the pathologist marked these areas on FFPE blocks, and took tissue cores from the marked fields for DNA isolation. DNA samples were amplified by Sanger sequencing. As controls for the LOH study, we used five germline DNA samples clearly heterozygous for the founder *GEN1* variant, which were isolated from blood samples of the same *GEN1* variant carriers, whose tissues were tested for LOH.

Given that a series of tissues, including >80% cancer cells, were chosen, we expect that complete loss of the wild-type allele would lead to about 80% reduction of the signal from the normal allele on sequencing chromatograms at the site of the small deletion. Therefore, to evaluate whether the wild-type allele is lost in cancer tissues, we calculated the ratio between the peak height of the first deleted nucleotide-nucleotide G (investigated peak) corresponding to the wild-type allele (marked by “a” in Figure 1) and the height of the control peak corresponding to the first nucleotide G located 3 bp 5′ to the deletion (marked by “b” on Figure 1). For five germline DNA samples of heterozygous *GEN1* variant carriers (controls), the median peak ratio a/b was 0.45 (range 0.43 to 0.51). Complete LOH would lead to about an 80% reduction of the height of peak “a”, and is expected to have no impact on the height of peak “b” located 5′ to the small deletion, thus a complete loss of the normal allele in tumor is expected to result in a reduction of the a/b peak ratio from approximately 0.45 (seen in heterozygous controls) to about 0.1.

Of note, Sanger sequencing is not an optimal technique for LOH analysis. In contrast to NGS, it does not provide exact data on the number of copies of each allele (mutant and wild-type). However, it is sufficient to provide information if the wild-type allele is lost by analyzing the sequencing peak height of a nucleotide corresponding to a normal allele at the site of a mutation (i.e., peak height for nucleotide G) versus the control peak height for the same type of nucleotide (nucleotide G), but localized 5′ and close to the variant (in the normal sequence).

The proportion of peak heights for the same type of nucleotide is similar in different samples, in particular when the nucleotides are localized close to Sanger sequencing chromatograms. Therefore, our LOH study was based on the analysis of the signals for the two closest nucleotides G (3 bp distance between the two nucleotides). Of note, extensive small deletions/insertions could lead to preferential amplification of a shorter allele in PCR. However, the *GEN1* deletion is small, including only 4 bp; therefore, it is unlikely that the PCR fragment with the deletion of 88 bp is more strongly amplified in the PCR reaction than the wild-type fragment of 92 bp. Another possible source of spurious results of LOH analysis is that DNA may be isolated from the tumor area, containing a low percentage of cancer cells compared to normal cells. We used an experienced pathologist in micro-dissection to minimize the chance of this error. It is also possible that DNA isolated from the tumor may be contaminated with normal DNA, leading to false-negative results of the LOH study. However, to avoid such a scenario, negative controls without DNA were used in all PCR reactions. The sample of the available five tumors was relatively small, which reduces the chance of identifying LOH. However, if *GEN1* acted as a classical and high- or moderate-risk tumor suppressor gene for breast cancer, we would expect to see LOH in most of the studied tumors. The existence of LOH could support a detected association, and in our case, when we did not find any association, it did not affect the results.

#### 4.2.4. Statistical Analysis

The prevalence of two recurrent *GEN1* variants was first estimated in 617 HBC cases and 300 controls (detected in the discovery sequencing phase). The prevalence of the c.1929_1932delAAAG allele was also estimated in 15,930 breast cancer cases and 4702 cancer-free women (genotyped in the validation phase of the study). Odds ratios were generated from two-by-two tables. Statistical significance was assessed using the Fisher exact test or the Chi-squared test, where appropriate. In addition, the adjusted Mantel–Haenszel odds ratio for unselected breast cancer given the c.1929_1932delAAAG allele was calculated stratified by five geographical regions of Poland (data for the five geographical regions were considered as five separate studies). The Breslow–Day test of homogeneity of odds ratios was used to calculate if there is a statistically significant difference between the odds ratios for the five geographical regions (Appendix A).

Women with breast cancer, with and without a GEN1 variant, were compared for age at diagnosis and other clinical features of their breast cancers. Statistical significance was assessed using the Fisher exact test or the Chi-squared test, where appropriate. Means were compared using a *t*-test. Medians were compared using the Mann-Whitney test. To estimate the survival of women with and without the c.1929_1932delAAAG variant, we followed the breast cancer patients from the date of diagnosis until the date of death or July 2014. Median follow-up between carriers and non-carriers was compared using the Mann-Whitney test. To estimate 5- and 10-year survival, Kaplan-Meier curves were constructed for variant carriers and non-carriers. The HR for mortality and significance given the c.1929_1932delAAAG variant was calculated using Cox regression analysis. All analyses were performed using MedCalc Version 23.2.1.

## 5. Conclusions

Our study and other studies suggest that *GEN1* variants do not confer an elevated risk of breast cancer (and probably other cancers). This data does not support the hypothesis that *GEN1* is a cancer susceptibility gene. Our study has clinical implications for genetic counseling and suggests the reclassification of *GEN1* changes as variants of uncertain significance (VUS) when they are detected in clinical testing panels.

## Figures and Tables

**Figure 1 ijms-26-05991-f001:**
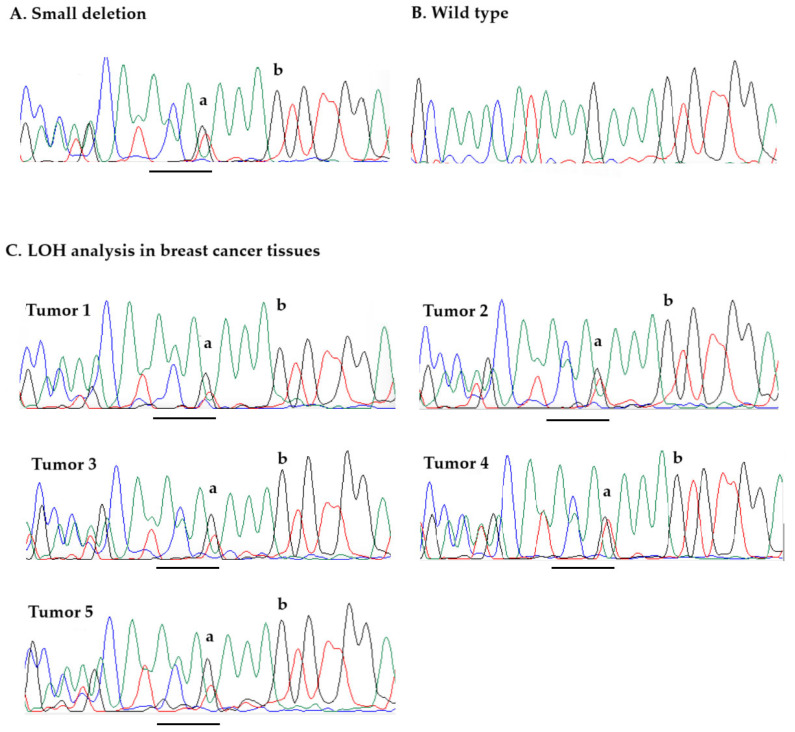
Sequencing chromatograms (sequencing results with the reverse primer are shown): (**A**). *GEN1* p.Lys645Cysfs*29 variant (example of sequencing of germline DNA from deletion carrier); (**B**). Wild-type sequence (example of sequencing of germline DNA from a non-carrier) (**C**). Loss of heterozygosity (LOH) analysis in breast cancer tissues from five carriers of p.Lys645Cysfs*29; retention of the GEN1 wild-type allele is observed in all cancers (peaks corresponding to nucleotides G, used in loss of the wild-type analysis (to produce a/b ratio) are presented by letters “a” and “b”). A 4-bp deletion of GEN1 is underlined. On sequence traces, green peaks correspond to nucleotide A, red to nucleotide T, black to nucleotide G, and blue to nucleotide C.

**Figure 2 ijms-26-05991-f002:**
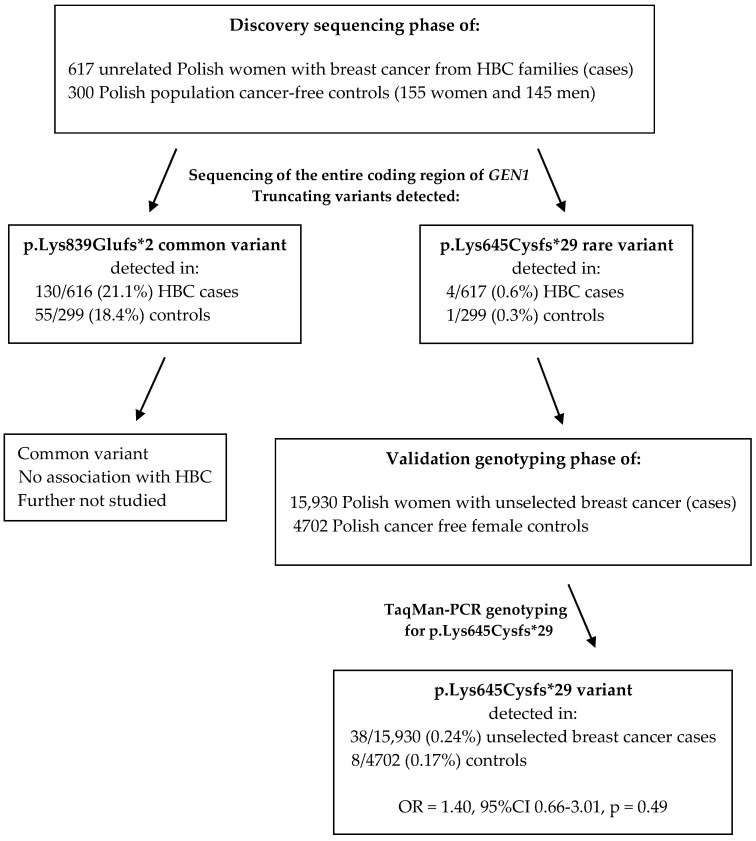
The flowchart describing the sequential approach used in the study.

**Table 1 ijms-26-05991-t001:** GEN1 truncating variants identified by sequencing in Polish patients with breast cancer from HBC families and in controls.

Variant ^a^	Exon	Genotype ^b^	Frequency Among Cases	Frequency Among Controls	*p* Value
c.1929_1932delAAAG (p.Lys645Cysfs*29)	13	del/wt	4/617 (0.6%)	1/299 (0.3%)	1.00
c.2515_2519delAAGTT (p.Lys839Glufs*2)	13	del/del	5/616 (0.8%)	4/299 (1.3%)	0.48
		del/wt	125/616 (20.3%)	51/299 (17.1%)	0.28
		del/del or del/wt	130/616 (21.1%)	55/299 (18.4%)	0.38

^a^ Nucleotide positions based on the NM_182625.4 transcript of *GEN1*; ^b^ del—deletion, wt—wild-type.

**Table 2 ijms-26-05991-t002:** Prevalence of the GEN1 c.1929_1932delAAAG (p.Lys645Cysfs*29) deletion in 15,930 women with unselected breast cancer, by age and family history, and in 4702 cancer-free women.

	Total (n)	GEN1 Variant-Positive	Prevalence (%)	OR (CI 95%)	*p*-Value
Patients with Breast Cancer
All cases	15,930	38	0.24%	1.40 (0.66–3.01)	*p* = 0.49
Age (years)
≤50	5723	9	0.16%	0.92 (0.36–2.40)	*p* = 0.87
51–60	4909	15	0.30%	1.80 (0.76–4.25)	*p* = 0.25
>60	5298	14	0.26%	1.55 (0.65–3.71)	*p* = 0.43
Number of Relatives with Breast Cancer *
0	14,592	28	0.19%	1.13 (0.51–2.48)	*p* = 0.92
1	1797	5	0.28%	1.64 (0.53–5.01)	*p* = 0.57
≥2	510	2	0.39%	2.31 (0.49–10.91)	*p* = 0.58

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

Ductal, grade unknown	2/32 (6.2%)	1014/12,452 (8.1%)	*p* = 0.70
Medullary	2/32 (6.2%)	259/12,452 (2.1%)	*p* = 0.12
Lobular	2/32 (6.2%)	1528/12,452 (12.3%)	*p* = 0.31
Tubulolobular	1/32 (3.1%)	263/12,452 (2.1%)	*p* = 0.69
DCIS	2/32 (6.2%)	475/12,452 (3.8%)	*p* = 0.48
Other or undefined	3/32 (9.4%)	1165/12,452 (9.4%)	*p* = 1.00
Receptor status:			
Estrogen receptor-positive	17/23 (73.9%)	7485/10,366 (72.2%)	*p* = 0.86
Progesterone receptor-positive	18/23 (78.3%)	7353/10,221 (71.9%)	*p* = 0.50
HER2-positive	3/19 (15.8%)	1601/8619 (18.6%)	*p* = 0.76
Size (cm):
<1	5/22 (22.7%)	1081/9966 (10.8%)	*p* = 0.08
1–1.9	8/22 (36.4%)	3999/9966 (40.1%)	*p* = 0.72
2–4.9	7/22 (31.8%)	4447/9966 (44.6%)	*p* = 0.23
≥5	2/22 (9.1%)	439/9966 (4.4%)	*p* = 0.30
Lymph node-positive	8/22 (36.4%)	4547/10,332 (44.0%)	*p* = 0.47
Bilateral	2/28 (7.1%)	496/11,755 (4.2%)	*p* = 0.45
Chemotherapy (yes)	10/25 (40.0%)	4631/8678 (53.4%)	*p* = 0.19
Vital status (deceased)	5/30 (16.7%)	2036/12,037 (16.9%)	*p* = 0.97

Data represent mean (range) or number/total (%), *p*-values compare variant-positive to variant-negative patients and were calculated using Fisher’s exact test.

**Table 4 ijms-26-05991-t004:** Cancers reported in the families of the 35 unselected cases of breast cancer with the GEN1 p.Lys645Cysfs*29 variant and in the 14,592 non-carrier patients.

Cancer Site	Number (%) of Cancers in Relatives of GEN1 Variant-Positive Patients (n = 35 Families)		Number (%) of Cancers in Relatives of GEN1 Variant-Negative Patients (n = 14,592 Families)		*p*-Value
	**N**	**%**	**N**	**%**	
Breast	7	20.0%	2431	16.7%	0.76
Colon	2	5.7%	1112	7.6%	0.92
Kidney	0	0.0%	411	2.8%	0.62
Larynx	2	5.7%	554	3.8%	0.88
Lung	5	14.3%	2156	14.8%	0.93
Leukemia or Lymphoma	1	2.9%	561	3.8%	0.76
Pancreas	0	0.0%	402	2.8%	0.63
Prostate	1	2.9%	991	6.8%	0.57
Stomach	2	5.7%	1243	8.5%	0.77
Cervix/endometrium	3	8.6%	1513	10.4%	0.94
Ovary	2	5.7%	538	3.7%	0.85

## Data Availability

The main research data supporting the results of this study are included in Table 1, Table 2, Table 3 and Table 4, Figure 1 and Figure 2, and Appendix A. Other data could be shared from the authors upon request.

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
