# Peer review of "Analysis of GEN1 as a Breast Cancer Susceptibility Gene in Polish Women"

_ijms, 2025, doi:10.3390/ijms26135991_

Round 1
Reviewer 1 Report
Comments and Suggestions for Authors
Gliniewicz et al. submitted an interesting original article investigating whether the GEN1 gene, involved in DNA damage repair, acts as a breast cancer susceptibility gene. This is a well-designed and thoroughly executed study that provides compelling evidence that GEN1 truncating mutations, particularly p.Lys645Cysfs*29, are not associated with increased breast cancer risk. The manuscript is very interesting, and it is a relevant and timely contribution, particularly for refining genetic screening strategies. However, I have some minor concerns that might be taken into consideration prior to publication.
1. Title: I would consider adding "in Polish Women" or "in a Population-Based Cohort" to strengthen the focus.
2. Abstract: I suggest specifying the population in which the study was conducted (e.g., Polish women) and including sample sizes of both hereditary and unselected breast cancer cohorts.
Strengths: Well-structured and effectively summarizes the aim, methods, findings, and conclusion.
3. Introduction: I believe that authors can elaborate more on the functional importance of GEN1 in homologous recombination with literature support (e.g., Ip et al., 2008; Rass et al., 2010). Also, I would like authors to consider stating why prior studies may have yielded inconclusive or contradictory results.
4. Methods: I have several suggestions in this section as follows:
- Authors should provide more details about the depth and quality of sequencing (e.g., mean coverage, percentage of target bases covered >20x).
- It would be better to clarify how variants were annotated and prioritized (e.g., reference databases used, in silico prediction tools).
- Also, I really suggest clearly differentiating between the use of sequencing vs. genotyping cohorts and provide justification for the sample selection strategy.
- It seems that the LOH methodology is underexplained. Could authors add controls used, threshold for LOH interpretation, and potential technical limitations?
5. Results: Regarding the statistical results (ORs and p-values) are presented but not fully interpreted. For example, although OR = 1.40 for p.Lys645Cysfs*29, the wide CI (0.66–3.01) and non-significance (p = 0.49) should be clearly stated as indicating no strong association. Also, Table 3 is informative, but statistical power may be low due to small numbers of mutation carriers. Please acknowledge this limitation.
6. Discussion: Authors should tell why truncating mutations in the C-terminal region (as opposed to the N-terminal catalytic domain) may not impact GEN1 function (see Ip et al., Nature, 2008; Rass et al., Genes Dev, 2010). Also, they should address the clinical implications of this study for genetic counseling and whether reclassification of GEN1 as a variant of uncertain significance (VUS) may be appropriate in clinical panels. Here is an important to discuss whether any functional studies are warranted to validate the impact of p.Lys645Cysfs*29 on protein function.
Line 177: Rephrase this phrase “The variant was much more common in Poland…” to “The variant appeared more frequently in our Polish cohort…”.
Author Response
We thank the Reviewer 1 for the thoughtful and constructive feedback. Below, we respond to each point raised by the Reviewer.
- Title: I would consider adding "in Polish Women" or "in a Population-Based Cohort" to strengthen the focus.
Response: We have updated the title to reflect the study population better. The new title is: "Analysis of GEN1 as a Breast Cancer Susceptibility Gene in Polish Women".
- Abstract: I suggest specifying the population in which the study was conducted (e.g., Polish women) and including sample sizes of both hereditary and unselected breast cancer cohorts.
Strengths: Well-structured and effectively summarizes the aim, methods, findings, and conclusion.
Response: We have included suggested information in the abstract (lines 18-39).
- Introduction: I believe that authors can elaborate more on the functional importance of GEN1 in homologous recombination with literature support (e.g., Ip et al., 2008; Rass et al., 2010). Also, I would like authors to consider stating why prior studies may have yielded inconclusive or contradictory results.
Response: We have now described the functional importance of GEN1 in more details (lines 78-94 and lines 259-268)
- Methods: I have several suggestions in this section as follows:
- Authors should provide more details about the depth and quality of sequencing (e.g., mean coverage, percentage of target bases covered >20x).
- It would be better to clarify how variants were annotated and prioritized (e.g., reference databases used, in silico prediction tools).
Response: The quality of sequencing was good. The mean depth of coverage was approximately 100x (range 52x to 154x) for the CCDS exons. On average 97.4% (range 91.2% to 99.1%) of the CCDS exons were covered at 20x depth of coverage or higher, and was used for variant calling. For GEN1, on average 99.8% of its coding exons (exons 2-14, NM_182625.4) was covered at 20x or more.
The SNP & Variation Suite (GoldenHelix Inc., Bozeman, MT, USA) was used for annotating called variants and to determine the effect of GEN1 variants on the coding protein. We searched for GEN1 loss-of-function variants, including frameshift indels, nonsense mutations, splicing site variants, and start codon loss mutations. We have now described WES in detail (lines 427-456)
- Also, I really suggest clearly differentiating between the use of sequencing vs. genotyping cohorts and provide justification for the sample selection strategy.
Response: Now we have better described our approach, clearly separated our discovery sequencing phase from the validation genotyping phase and have provided justification for the sample selection strategy (new text, Material and Methods, lines 327-356).
- - It seems that the LOH methodology is underexplained. Could authors add controls used, threshold for LOH interpretation, and potential technical limitations?
Response: Now we have better described our LOH study, limitations (new text, Material and Methods, lines 485-530) and its results (Results, lines 176-190).
- Results: The statistical results (ORs and p-values) are presented but not fully interpreted. For example, although OR = 1.40 for p.Lys645Cysfs*29, the wide CI (0.66–3.01) and non-significance (p = 0.49) should be clearly stated as indicating no strong association. Also, Table 3 is informative, but statistical power may be low due to small numbers of mutation carriers. Please acknowledge this limitation.
Response: We have improved the Results section by acknowledging limitations of our results due to small number of mutation carriers (lines 135-138 and lines 154-155).
- Discussion: Authors should tell why truncating mutations in the C-terminal region (as opposed to the N-terminal catalytic domain) may not impact GEN1 function (see Ip et al., Nature, 2008; Rass et al., Genes Dev, 2010). Also, they should address the clinical implications of this study for genetic counseling and whether reclassification of GEN1 as a variant of uncertain significance (VUS) may be appropriate in clinical panels. Here is an important to discuss whether any functional studies are warranted to validate the impact of p.Lys645Cysfs*29 on protein function.
Response: We thank the Reviewer for these insightful comments. Suggested changes are now included in the revised Discussion section (lines 259-268, lines 276-279) and we have modified our conclusions (last lines 35-39 of the Abstract, lines 320-323 of the Discussion section, and the Conclusions section lines 558-560). - Line 177: Rephrase this phrase “The variant was much more common in Poland…” to “The variant appeared more frequently in our Polish cohort…”.
Response: We have rephrased the sentence accordingly (The Discussion, line 233).

Reviewer 2 Report
Comments and Suggestions for Authors
The manuscript looks interesting, but some modifications are recommended:
- The title is very short and not inclusive or comprehensive. You should modify the title to be more informative.
- In introduction: I suggest adding the last GLOBOCAN breast cancer statistics (2022) in the first paragraph.
- Throughout the manuscript, the authors must write the full name of any abbreviation in its first mention.
- In line 54 and 55 "Most breast cancer susceptibility genes are involved in the damage repair signaling 54
pathway, i.e. including BRCA1, BRCA2, PALB2, CHEK2, ATM, BARD1, RAD51C, and RAD51D", I'm wondering you did not mention GEN1 in this list and mention it later, why? Is GEN1 less important or not common as the others? - "Materials and Methods" must be changed to "patients and methods" and 4.1 materials should be patients or participants.
- On which basis the authors determined their sample size to be 617 breast cancer cases the free controls? is that sample size is statistically significant to generalize the findings?
- You should change selected and unselected cases to be inclusion and exclusion criteria and rewrite as criteria.
- You should clarify that your study is multicentred; you recruited patients from different medical centre.
- Why you included 145 men as controls? the controls should be matching as much as possible, and your BC cases were all women!!!
- I suggest adding flowchart for the screened and the finally involved patients and controls because too much details and numbers are there, which may make readers to be confused.
- You included 617 BC cases in your study, but performed Loss of Heterozygosity Analysis only on 6 samples, which were obtained from patients other than those 617?!! I find this is not acceptable. Why you did not obtain the samples for pathology examination from your already involved patients? or why did not contact those 6 to be with others 617?!
- I suggest making ROC curve analysis and determine the sensitivity and specificity for GEN1 to confirm your hypothesis.
- The authors must add table in their result about the demographic data of their included participants.
- The discussion lacks perspectives and solutions to confirm in other settings the results described in the manuscript. This must be included.
- You mentioned in the manuscript there is a suppletory materials, but no other files were uploaded.
- The plagiarism (43%) as appears in the system is considerably high, I suggest lowering this percent.
Author Response
We thank the reviewer for the thorough and constructive comments. We have carefully addressed each point as outlined below:
1. The title is very short and not inclusive or comprehensive. You should modify the title to be more informative.
Response: We have updated the title to better reflect the focus of the study. The revised title is: "Analysis of GEN1 as a Breast Cancer Susceptibility Gene in Polish Women”.
2. In introduction: I suggest adding the last GLOBOCAN breast cancer statistics (2022) in the first paragraph.
Response: We have added GLOBOCAN 2022 data to the Introduction (lines 47-49) and quoted the relevant reference (ref. 2).
3. Throughout the manuscript, the authors must write the full name of any abbreviation in its first mention.
Response: Thank you for pointing this out. We have carefully reviewed the entire manuscript and now have spelled out in full all abbreviation when they are first used.
4. In line 54 and 55 "Most breast cancer susceptibility genes are involved in the damage repair signaling 54 pathway, i.e. including BRCA1, BRCA2, PALB2, CHEK2, ATM, BARD1, RAD51C, and RAD51D", I'm wondering you did not mention GEN1 in this list and mention it later, why? Is GEN1 less important or not common as the others?
Response: The fact that GEN1 is described later than the other genes (in a separate paragraph) does not mean it is less important than the others. We did not list GEN1 together with other well established (high or moderate risk) breast cancer susceptibility genes that are involved in DNA repair, because there is no evidence that GEN1 is a breast cancer susceptibility gene. An analysis of the association of GEN1 with breast cancer susceptibility is the aim of our study, therefore intentionally GEN1 is mentioned later in a separate (dedicated to GEN1) paragraph, in which we described the gene and its function in detail, and provided clear rationale that it is a good candidate for breast cancer susceptibility gene.
5. "Materials and Methods" must be changed to "patients and methods" and 4.1 materials should be patients or participants.
Response: According to this suggestion we have changed “Materials and Methods” to “Patients and Methods” and “Material” to “Participants” (line 325 and 326)
6. On which basis the authors determined their sample size to be 617 breast cancer cases the free controls? is that sample size is statistically significant to generalize the findings?
GEN1 sequencing data were derived for a cohort of all available to us 617 Polish women with breast cancer from HBC families who previously were screened using exome sequencing (WES) by our group. So we selected all 617 Polish HBC families with a strong clustering of breast cancer (on average 3.4 breast cancer cases per family) for the study. In the discovery phase as a reference we used all available to us, 300 cancer-free Polish population controls who were screened by exome sequencing previously (as a consequence of our study “Polinome” performed to obtain 300 Polish exomes from cancer-free individuals that may be used as a reference for interpretation of pathogenicity of germline variants detected in Polish cancer patients).
Our study benefits from genetic homogeneity of Poland which is populated by ethnic Poles, and we restricted our study to individuals who self-reported themselves as ethnic Poles (of note, cases and controls were enrolled before full scale Russian-Ukrainian war in 2021, when Poland accepted above 1 million refugees from Ukraine). We don’t observe significant differences in the allele frequencies between different regions of Poland i.e. for BRCA1, PALB2, CHEK2, NBN, RECQL, and also GEN1 founder variants, neither in breast cancer cases nor in controls (for GEN1 data see Supplementary Table 1). In most well established breast cancer susceptibility genes, Polish common founder mutations were detected (i.e. BRCA1, BRCA2, PALB2, CHEK2), which constitute the majority (>80%) of all mutations detected in these genes in Polish women with breast cancer.
We studied individuals from a genetically homogenous population, so we searched for recurrent (founder) truncating variants by sequencing of 617 unrelated Polish HBC families. Using binominal distribution we estimate that if an allele is present in 3 of the 1000 Polish HBC families in a given gene, we have a power of 55% to detect this allele in two or more families. If the allele is present in 6 of 1000 or more of Polish HBC families, we have a power of 88% to detect this allele in two or more families. In fact, we identified two recurrent variants of GEN1, one common (p.Lys839Glufs*2) detected in 21% of HBC families, and the other rare variant (p.Lys645Cysfs*29) seen in 0.6% of Polish HBC families. Therefore, given analysis of the power to detect GEN1 variants in our discovery sequencing phase by binominal distribution (among 617 Polish HBC families), a large-scale multicenter association study in the genotyping validation phase which included about 16,000 unselected breast cancer cases and 5000 controls from all over Poland, and a genetic homogeneity of the Polish population, we consider that we have sufficient data and power to detect possible association between GEN1 and breast cancer susceptibility.
We have added above explanation to the Discussion (lines 280-305)
7. You should change selected and unselected cases to be inclusion and exclusion criteria and rewrite as criteria.
Response: We have now clearly described our cases and controls and how they were enrolled in the Patients and Methods section (including that HBC families with a strong clustering of breast cancer used in the discovery sequencing phase, and unselected breast cancer cases were used in the validation genotyping phase of the study). We have now also better described our approach, clearly separated our discovery sequencing phase from the validation genotyping phase and have provided justification for the sample selection strategy (as requested by the Reviewer 1) (new text, Patients and Methods, lines 327-356).
8. You should clarify that your study is multicentred; you recruited patients from different medical centre.
Response: We have included the requested information in the Participants section (lines 348-356).
Why you included 145 men as controls? the controls should be matching as much as possible, and your BC cases were all women!!!
Response: We acknowledge the reviewer’s concern regarding sex-matching in case-control studies. However, in the context of germline genetic association studies, particularly for pathogenic variants in high-penetrance cancer susceptibility genes, the underlying mutation frequency in the general population is not expected to differ substantially between males and females. This is especially true when the outcome of interest is carrier frequency, not disease expression, and the controls are used solely to estimate background allele frequency in the population. In fact, we saw no significant difference in the studied GEN1 variant frequencies (for both truncating GEN1 variants) by sex among the 300 controls (explained in lines 333-342).
Including males in the control group allows for a more accurate and stable estimate of population-level variant frequency, especially for rare variants. However, if the Editors feel it is necessary to restrict our controls in the discovery series to 155 women, we are happy to do so, but this effort will have no impact on the outcome of the discovery sequencing phase.
9. I suggest adding flowchart for the screened and the finally involved patients and controls because too much details and numbers are there, which may make readers to be confused.
Response: Thank you for your suggestion. We have added the flowchart showing the structure of our sequential approach (Patients and Methods, Figure 2, between lines 357 and 358).
10. You included 617 BC cases in your study, but performed Loss of Heterozygosity Analysis only on 6 samples, which were obtained from patients other than those 617?!! I find this is not acceptable. Why you did not obtain the samples for pathology examination from your already involved patients? or why did not contact those 6 to be with others 617?!
Response: We thank the reviewer for raising this important point. It is hard to get FFPE samples from most cases because majority of breast cancers were diagnosed before year 2010. We were able to get 6 tumors and to perform LOH analysis on 5 tumors (from GEN1 variant carriers) which were deposited in the Department of Pathology of Pomeranian Medical University in Szczecin (the pathologist associated with the study was employed in the Department). Now we have clarified that we were able to obtain tumor samples and to perform LOH for two GEN1 deletion carriers identified by WES among 617 patients with HBC, and three carriers identified by genotyping of a series of 15,930 unselected breast cancer cases (lines 475-478).
We are aware that the sample of available five tumors is relatively small that reduces the chance of identifying LOH. However, existence of LOH could support a detected association, and in our case when we did not find any association, it does not affect the results.
11. I suggest making ROC curve analysis and determine the sensitivity and specificity for GEN1 to confirm your hypothesis.
Response: Thank you for this suggestion. We appreciate the Reviewer’s interest in fully characterizing the data. However, we respectfully believe that a receiver-operating-characteristic (ROC) analysis is not informative for the present study, for three reasons:
- Purpose of ROC curves.
ROC curves quantify the diagnostic classification performance of a marker across varying thresholds, yielding sensitivity, specificity and area-under-the-curve (AUC). They are appropriate when the goal is to decide, for an individual, whether the marker can predict disease status.
b. Study objective and data structure.
Our study is an association analysis in a case–control framework. We test whether the frequency of pathogenic or likely pathogenic GEN1 variants differs between two groups at the population level. The relevant effect estimate is an odds ratio (OR) with its 95 % CI, obtained by Fisher’s exact test; this is what we have reported.
c. Lack of a discriminatory signal.
Because mutation frequencies are virtually similar in cases and controls (OR ≈ 1), any ROC curve would have a single point on the diagonal (sensitivity = 1-specificity), producing an AUC of ~0.50. This confirms there is no diagnostic utility, but adds no information beyond the association statistics already provided. For these reasons we have not included a ROC analysis.
If the Editors nevertheless feel it would be helpful for completeness, we can add a supplementary figure showing the ROC curve (AUC ~0.50) and a brief statement that it corroborates the null association.
We hope this explanation clarifies our reasoning, and we remain happy to provide additional details if required.
12. The authors must add table in their result about the demographic data of their included participants.
Response: Poland is populated by ethnic Poles and it is well known founder and genetically homogenous population. Therefore, it is perhaps not surprising that GEN1 c.1929_1932delAAAG (p.Lys645Cysfs*29) variant frequency is similar in different regions of Poland (see new Supplementary Table 1).
Only Polish cases and controls were enrolled to our study (all enrolled individuals reported that their all grandparents are of Polish ethnicity) (new text, Patients and Methods, lines 349-355). Therefore, women with breast cancer and cancer free controls are matched by ethnicity, the most important factor for the association study.
13. The discussion lacks perspectives and solutions to confirm in other settings the results described in the manuscript. This must be included.
Response: We have now significantly extended the Discussion according to the Reviewers’ comments. We have discussed our results and other reports that analyzed the association of GEN1 and breast cancer. We suggested that GEN1 variants are uncommon and further epidemiological and functional studies are needed to explore if GEN1 mutations may coffer increase in the risk of breast cancer. We have also improved our conclusions and explained that our analysis has clinical implications. We hope that now the Reviewer 2 and the Editors find our discussion satisfactory.
14. You mentioned in the manuscript there is a suppletory materials, but no other files were uploaded.
Response: We have now uploaded Supplementary Table 1 to present geographical distribution of the GEN1 c.1929_1932delAAAG (p.Lys645Cysfs*29) deletion among 15,930 women with unselected breast cancer, and among 4702 cancer-free female controls
15. The plagiarism (43%) as appears in the system is considerably high, I suggest lowering this percent.
Response: We understand the concern regarding the similarity index. The plagiarism is 27% according to MDPI: 7% MDPI-res.com, 7% www.mdpi.com, and 1% or below for several other sources. The Reviewer states that the plagiarism is 43%. It is likely the Reviewer 2 included the Patients and Methods section in the plagiarism analysis. We are convinced that this section should be excluded from the similarity report, because we used similar groups of cases and controls in our previous papers in which we analyzed association between different genes and breast cancer risk.
We have carefully reviewed MDPI similarity report and found that the similarity index does not include the entire paragraphs and sentences in the main text. The majority of overlapping content consists of standard phrases commonly used in scientific writing and terminology used in genetic cancer research. Additionally, a significant portion of the overlap is due to institutional information, ethical declarations, and standardized journal formatting provided by MDPI (e.g., authors, affiliations, funding statements, informed patient consent).
We have now significantly revised the paper, added new paragraphs, prof. Steven Narod corrected English language and rephrased the text, reducing the plagiarism. We also confirm that the manuscript is original, and all sources are appropriately cited.

Reviewer 3 Report
Comments and Suggestions for Authors
This is an overall well written paper that studies the possibl role of GEN1 variants in the pathogenesis of breast cancer. It focuses on two variants, one is a previously described variant that is considered not pathogenic. The other is a variant that has been the subject of debate in the literature, although it is already reported by CLINVAR as a VUS. The authors conclude that also this variant has no role in breast cancer.
The paper has some interest for people involved in designing genetic panels for hereditary brest cancer and should probably be published.
I would recommend not to use the term mutation when describing both the variants in the text. The authors repeatedly define the variants as mutation although they deny any role of the variants in the development of hereditary breast cancer. The term mutation should not be used
Author Response
- This is an overall well written paper that studies the possible role of GEN1 variants in the pathogenesis of breast cancer. It focuses on two variants, one is a previously described variant that is considered not pathogenic. The other is a variant that has been the subject of debate in the literature, although it is already reported by CLINVAR as a VUS. The authors conclude that also this variant has no role in breast cancer.
Response: We thank the reviewer for the positive assessment of our manuscript and for highlighting the relevance of our study design and conclusions. We appreciate the recognition of the clinical importance of clarifying the role of GEN1 variants in breast cancer pathogenesis.
- The paper has some interest for people involved in designing genetic panels for hereditary breast cancer and should probably be published.
Response: We thank the reviewer for this encouraging feedback.
- I would recommend not to use the term mutation when describing both the variants in the text. The authors repeatedly define the variants as mutation although they deny any role of the variants in the development of hereditary breast cancer. The term mutation should not be used.
Response: We fully agree with the reviewer that terminology is critical, particularly in clinical genetics. We have revised the manuscript to consistently use the term “variant” instead of “mutation” when referring to the two GEN1 changes analyzed in our study.

Round 2
Reviewer 2 Report
Comments and Suggestions for Authors
Thank you for your effort to enhance the manuscript. All comments have been covered.